# Evaluation of Chemical and Physical Triggers for Enhanced Photosynthetic Glycerol Production in Different *Dunaliella* Isolates

**DOI:** 10.3390/microorganisms12071318

**Published:** 2024-06-28

**Authors:** Linda Keil, Farah Mitry Qoura, Jonas Martin Breitsameter, Bernhard Rieger, Daniel Garbe, Thomas Bartholomäus Brück

**Affiliations:** 1Werner Siemens Laboratory of Synthetic Biotechnology, TUM-School of Natural Sciences, Technical University of Munich (TUM), 85748 Garching, Germany; linda.keil@tum.de; 2Fraunhofer Institute for Interfacial Engineering and Biotechnology IGB, Nobelstraße 12, 70569 Stuttgart, Germany; farah.mitry.qoura@igb.fraunhofer.de; 3Wacker-Laboratory of Macromolecular Chemistry, TUM-School of Natural Sciences, Technical University of Munich (TUM), 85748 Garching, Germany; jonas.breitsameter@tum.de (J.M.B.); rieger@tum.de (B.R.)

**Keywords:** *Dunaliella*, glycerol production, optimization, phylogenetic analysis, *Dunaliella* biodiversity, new strains

## Abstract

The salt-tolerant marine microalgae *Dunaliella tertiolecta* is reported to generate significant amounts of intracellular glycerol as an osmoprotectant under high salt conditions. This study highlights the phylogenetic distribution and comparative glycerol biosynthesis of seven new *Dunaliella* isolates compared to a *D. tertiolecta* reference strain. Phylogenetic analysis indicates that all *Dunaliella* isolates are newly discovered and do not relate to the *D. tertiolecta* reference. Several studies have identified light color and intensity and salt concentration alone as the most inducing factors impacting glycerol productivity. This study aims to optimize glycerol production by investigating these described factors singularly and in combination to improve the glycerol product titer. Glycerol production data indicate that cultivation with white light of an intensity between 500 and 2000 μmol m^−2^ s^−1^ as opposed to 100 μmol m^−2^ s^−1^ achieves higher biomass and thereby higher glycerol titers for all our tested *Dunaliella* strains. Moreover, applying higher light intensity in a cultivation of 1.5 M NaCl and an increase to 3 M NaCl resulted in hyperosmotic stress conditions, providing the highest glycerol titer. Under these optimal light intensity and salt conditions, the glycerol titer of *D. tertiolecta* could be doubled to 0.79 mg mL^−1^ in comparison to 100 μmol m^−2^ s^−1^ and salt stress to 2 M NaCl, and was higher compared to singularly optimized conditions. Furthermore, under the same conditions, glycerol extracts from new *Dunaliella* isolates did provide up to 0.94 mg mL^−1^. This highly pure algae-glycerol obtained under optimal production conditions can find widespread applications, e.g., in the pharmaceutical industry or the production of sustainable carbon fibers.

## 1. Introduction

The genus *Dunaliella* falls under the well-characterized *Dunaliellaceae* family. This family includes four sections: *Tertiolectae*, *Dunaliella*, *Viridis*, and *Peirceinae*, each with distinct characteristics related to salinity and carotene accumulation [1,2]. Species assigned to these four sections are found in diverse marine environments, including oceans, brine lakes, salt marshes, salt lagoons, and adjacent saltwater areas [3]. Section *Tertiolectae* comprises oligo-euhaline species that do not accumulate carotenes and thrive at an optimal salinity of <6% *w*/*w* NaCl. Section *Dunaliella* consists of halophilic species that accumulate carotenes. Notably, the *Dunaliella* section encompasses species like *D. salina*, *D. parva*, *D. pseudosalina*, and *D. bardawil*, with ongoing debates about the latter’s classification [4]. Section *Viridis* includes hypersaline species that are consistently green and radially symmetrical. *Peirceinae*, the final section, features hypersaline species that are always green, but their cells exhibit bilateral symmetry [2]. Although this classification appears clear, a study from 2006 discussed that the taxonomy of the green algae genus *Dunaliella* is often perceived as confusing. The names linked to species in culture collections should, at times, be approached with caution [5]. In those circumstances, this study categorized various *Dunaliella* isolates obtained from environmental samples within the *Dunaliella* family. They will be phylogenetically classified, and the isolate’s capability for intracellular glycerol production will be assessed.

Generally, all reported *Dunaliella* strains are very salt-tolerant, and some are able to grow with salt concentrations up to saturation of about 5.5 M NaCl [6]. The cells lack a rigid polysaccharide cell wall and only possess a flexible cytoplasmic membrane [7]. This allows the algae to quickly adapt to hyperosmotic changes by reducing the size of the cell, followed by glycerol production either via photosynthetic CO_2_ fixation, or starch degradation [8,9]. The generated glycerol serves as an osmoprotectant to avoid cell bursts [7]. The level of intracellular glycerol is proportional to the extracellular salt concentration, reaching values higher than 50% of the dry weight of the cell [10]. 

Glycerol is an essential raw material to produce various valuable products such as pharmaceuticals, food, cosmetics, paints, and textiles [11,12,13,14]. The glycerol market size reached over USD 2 billion in 2021 and is expected to increase annually by over 7% from 2022 to 2030 [15]. Glycerol will be an important compound for high-value chemicals and biopolymers in the future [16]. As a promising precursor for ‘green chemicals’, glycerol has the chance to replace several fossil-based chemicals such as ethylene and propylene glycols, 2,3-butanediol, and acrolein [16,17,18,19,20]. Additionally, glycerol constitutes as a precursor for polyacrylonitrile (PAN), which is one of the most important polymers used in industry [21]. PAN in turn is the precursor for many products, such as resins, plastics, and acrylic fibers [22,23], as well as carbon fibers [24,25]. (Poly-) Acrylonitrile is synthetized via the SOHIO process [26], which is a fossil-based process via the ammoxidation of propene [27] and comes along with high carbon dioxide emissions. A less CO_2_ emitting process is the conversion of glycerol to acrylonitrile, especially if bio-based glycerol serves as a raw material [23].

Large demands for bio-based glycerol are met through biodiesel production, which generates glycerol as a by-product [23,28]. However, beyond the high prices and availability fluctuations [29], there are many impurities in biodiesel [29], which raise the purification costs as complex purification processes are needed [30,31]. Biodiesel production utilizes enormous agricultural areas necessary to nourish the world population [32,33,34]. Therefore, a sustainable and less area-intensive production process for the extensively required raw material, glycerol, is of great importance [28,35]. Used cooking oil might be a sustainable alternative for biodiesel and glycerol production [36], but it cannot satisfy market volumes [32,37]. 

Microalgae are an appropriate alternative that allows for the environmentally friendly synthesis of glycerol [23], as they are unicellular aquatic microorganisms that perform photosynthesis and thereby convert sunlight, the greenhouse gas CO_2_, and inorganic nutrient salts into biomass and glycerol [38,39,40,41]. Compared to terrestrial plants, microalgae grow up to ten times faster [42,43] and photosynthesize about ten times as efficiently as plants do, owing to their energy-efficient photosynthetic apparatus [44,45]. Moreover, it is feasible to cultivate microalgae on wasteland using saline or waste water [46,47] with the result that they do not compete with agricultural food production. These advantages make microalgae a suitable biotechnological tool to efficiently fix CO_2_ and convert it into sustainable, high-value products [48,49].

This paper aims to optimize the overall glycerol titer extracted from algae with respect to potential future applications, such as the pharmaceutical industry or the production of green chemicals. Different *Dunaliella* isolates from environmental samples, were examined with respect to their capacity for glycerol biosynthesis in comparison to the well-studied *D. tertiolecta.* The most promising candidates were applied in a process to optimize parameters that impact glycerol synthesis, such as light and NaCl concentration [8,50,51]. Even though these are already known factors, Xu et al. [52] have demonstrated that each strain behaves differently. Therefore, we applied those factors to our strains to investigate them individually. Moreover, the optimized conditions were combined to further improve the glycerol titer, as these factors have only been studied alone and not in combination. In synergy with the obtained biochemical data, the new *Dunaliella* isolates were subjected to a phylogenetic analysis, which indicated that all strains are newly discovered and are not related to the *D. tertiolecta* reference despite their capacity to biosynthesize glycerol in the absence of ß-carotene formation.

## 2. Materials and Methods

To provide a better overview of the experiments conducted in this study, a flowchart (Figure 1) summarizes the main steps. These steps are described in more detail in the following sections.

### 2.1. Algae Strains

*D. tertiolecta* UTEX 999 was obtained from the Culture Collection of Algae at the Georg August University of Göttingen. The strains *Dunaliella* sp. 6, *Dunaliella* sp. 27, *Dunaliella* sp. 37, *Dunaliella* sp. 83, *Dunaliella* sp. 96, *Dunaliella* sp. 101, *Dunaliella* sp. 127 (in figures named #6, #27, #37, #83, #96, #101, and #127, respectively) were isolated from environmental samples obtained from Australian saltwater lakes in August 2014 by Thomas Brück and coworkers [50]. The corresponding coordinates are specified in the Appendix A. The purity of the derived unialgal cultures was verified by microscopy (Zeiss AxioLab A.1, Carl Zeiss AG, Oberkochen, Germany). 

### 2.2. Microalgae Cultivation

All microalgae strains were cultured in 500 mL Erlenmeyer flasks with a culture volume of 200 mL in New Brunswick Innova 44 series shakers (28 °C, 120 rpm) fitted with light-emitting diodes (LED) (FUTURELED GmbH, Berlin, Germany), as described by Woortmann et al. [53]. If not otherwise mentioned, a continuous illumination at a Photosynthetic Photon Flux Density (PPFD) of 100 μmol m^−2^ s^−1^ between 400 nm and 750 nm was applied, to achieve a visible sunlight spectrum (AM1.5 Global). Each flask was aeriated with a rate of 16 sL h^−1^ with 2% *v*/*v* CO_2_ enriched air which was controlled by a DASGIP^®^ MX module (Eppendorf AG, Hamburg, Germany). The microalgae starting with an OD_750_ of 0.1 were cultivated for 14 days in modified Johnson medium (1.5 g L^−1^ MgCl_2_·6H_2_O, 0.5 g L^−1^ MgSO_4_·7H_2_O, 0.2 g L^−1^ KCl, 0.2 g L^−1^ CaCl_2_·2H_2_O, 1.0 g L^−1^ KNO_3_, 0.043 g L^−1^ NaHCO_3_, 35 mg L^−1^ KH_2_PO_4_, 1.89 mg L^−1^ Na_2_EDTA, 2.44 mg L^−1^ FeCl_3_·6H_2_O, 610 µg L^−1^ H_3_BO_3_, 380 µg L^−1^ (NH_4_)_6_Mo_7_O_24_·4H_2_O, 60 µg L^−1^ CuSO_2_·5H_2_O, 51 µg L^−1^ CoCl_2_·6H_2_O, 41 µg L^−1^ ZnCl_2_, 41 µg L^−1^ MnCl_2_·4H_2_O) at a pH of 7.5 [54] with 1 M NaCl, if not otherwise specified. 

### 2.3. Phylogenetic Characterization of Strains

DNA of the strains was extracted by the InnuPrep plant DNA extraction kit (Analytic Jena AG, Jena, Germany, 845-KS-1060050). The strains were identified as *Dunaliella* strains by the amplification of the 18S rDNA using the primers EukA (21F) (AACCTGGTTGATCCTGCCAGT) and EukB (1791R) (GATCCTTCTGCAGGTTCAC CTAC) [55]. The amplification occurred via an initial denaturation at 95 °C for 5 min followed by 35 cycles of 95 °C for 30 s, 59.8 °C for 30 s, and 72 °C for 1.5 min with a final extension step of 10 min at 72 °C. To further identify the species, the internal transcribed spacer (ITS)-sequence as well as the rubisco gene were amplified. Therefore, the ITS Primer ITS3 (GCATCGATGAAGAACGCAGC) and ITS4 (TCCTCCGCTTATTGATATGC) [56] and the rubisco primer rbcLaF (ATGTCACCACAAACAGAGACTAAAGC) and rbcLaR (GTAAAATCAAGTCCACCRCG) [57] were used. ITS sequence was amplified by applying an initial denaturation step at 95 °C for 5 min, followed by 35 cycles of 95 °C for 30 s, 54 °C for 30 s and 72 °C for 45 s. For the amplification of the rubisco gene, after initial denaturation at 98 °C for 2 min, 30 cycles of 98 °C for 10 s, 55 °C for 20 s, and 72 °C for 1 min was applied. The purified amplicons were sequenced by sanger sequencing (Eurofins Genomics GmbH, Ebersberg, Germany) and the obtained sequences were searched against the GenBank database by the BLASTn algorithm [58]. The 18S sequence was used for phylogenetic dendrogram design. Thus, the neighbor-joining method was applied by using the software from PHYLIP, version 3.57c (https://phylipweb.github.io/phylip/ (accessed on 12 June 2024)). The DNADIST program with Kimura-2 factor was used to compute the pairwise evolutionary distances for the above aligned sequences. The topology of the phylogenetic tree was evaluated by performing a bootstrap (algorithm version 3.6b) with 1000 bootstrapped trials. The tree was drawn using Tree View 32 software. 

### 2.4. Glycerol Extraction and Measurement

After 14 days of cultivation the NaCl concentration was abruptly increased, if not stated otherwise, to 2 M by adding the same volume of 3 M NaCl solution. As a control, the NaCl concentration was kept constant at 1 M NaCl, and instead of 3 M, the same volume of a 1 M NaCl solution was added. The culture was cultivated under the same conditions as before. To measure the accumulated glycerol, the biomass of *D. tertiolecta* and all isolates was harvested immediately after the addition of NaCl or after 24 h by centrifugation of 1 mL culture (2500× *g*, 5 min). The supernatant was removed. Half of the initial volume of bi-distilled H_2_O was added to the cell pellet. Cells were resuspended and centrifuged at 15,000× *g* for 5 min. After repeating the previous step, the glycerol content of the extract was determined using either the Glycerin-Assay-Kit (MAK117, Merck, Darmstadt, Germany) or the Free Glycerol Determination Kit (FG0100, Merck, Darmstadt, Germany) In addition to the manufacturer’s instructions, for the Free Glycerol Determination Kit (FG0100), sample and reagent volumes was halved. Modified Johnson medium was applied as a blank. 

### 2.5. Analyzing the Impact of Salt on Glycerol Production

Since glycerol serves as an osmoprotectant to the culture’s salt concentration, salt influence the glycerol titer [59]. To investigate the impact of the salt concentration, the algae were cultivated at different salt concentrations (1 M, 1.5 M, and 2 M) and after 14 days the NaCl concentration was raised to 2 M, 3 M, and 4 M by adding NaCl solutions in the required molarity. For the measurement of the glycerol concentration, the cells were harvested at 0 h and 24 h after salt stress. As a control the cultures were diluted with the same volume, but the salt concentration was retained constant.

### 2.6. Analyzing the Impact of Light on Glycerol Biosynthesis

Glycerol production potential was evaluated with the impact of light, which is a key component for the growth of autotrophic algae based on photosynthesis [40,41], the impact of light on the growth of the algae, as well as their glycerol production potential, was evaluated. For this purpose, the algae were cultivated with different light intensities and -colors. To analyze the impact of the light intensity, the *D. tertiolecta* and *Dunaliella* sp. 96 were cultivated with 100, 500, 1000, 1500, and 2000 μmol m^−2^ s^−1^ intensity of the visible sunlight spectrum (in the following called white light). Additionally, the effect of red and blue light on the algae growth and glycerol concentration was investigated. Thus, 100 and 1000 μmol m^−2^ s^−1^ were adjusted by using only light of the wavelengths 680 nm and 740 nm for red light conditions and 425 nm, 455 nm, and 470 nm for blue light.

### 2.7. Combining Improved Light and Salt Conditions to Increase Glycerol Production Titer

As a final experiment, the best conditions of the previous investigations were combined to optimize the glycerol titer. In addition to *D. tertiolecta* and *Dunaliella* sp. 96, the two isolates *Dunaliella* sp. 27 and *Dunaliella* sp. 83 were tested. Optimized cultures were cultivated at 1.5 M NaCl concentration with a light intensity of 500 μmol m^−2^ s^−1^. After 14 days of cultivation, salt was increased to 3 M NaCl and the cells were harvested 0 h and 24 h after increased salt concentration under same culture conditions. The non-optimized condition was cultivated as mentioned before. 

### 2.8. Nuclear Magnetic Resonance (NMR) Spectroscopy

For NMR analysis, glycerol was extracted with GC-grade ethanol. Therefore, the biomass was lyophilized (Zirbus Technology, Bad Grunz, Germany) before glycerol extraction. The ethanol extract was treated with activated carbon. After centrifugation at 15,000× *g* for 5 min, supernatant was lyophilized again, and algae derived glycerol remained. ^1^H-NMR spectra of algae extracted glycerol samples were recorded on a Bruker Ascend 400 MHz spectrometer. All spectra are referenced on the proton signal of CDCl_3_ (7.26 ppm).

### 2.9. Statistical Analysis

At least three biological replicates were measured for each experiment. In the figures, the results are expressed as mean ± standard deviation. A statistical analysis, comparing evaluated conditions with the control (100 μmol m^−2^ s^−1^, white light, cultivation at 1 M followed by an increase to 2 M NaCl) was performed via *t*-test, and the level of statistical significance was set to *p* < 0.05.

## 3. Results

By 18S-analysis (Appendix A), seven isolates from environmental samples were identified as *Dunaliella* strains. In addition, the analysis of the ITS sequence confirmed them as *Dunaliella* strains (Appendix A). The sequence of the rubisco gene analysis did cause ambiguous results, as the available database for algae was insufficient for a definitive taxonomic classification. 

To analyze the taxonomic relationship between the strains, for each strain, an NCBI blast alignment was performed. For each strain, a tree with 10 strains that had the highest identity compared to its own sequence was designed. For the final tree (Figure 2), these results of the isolates’ strains were fused, excluding any redundant data. The analysis highlights that all seven isolates can be considered new species, as indicated by a high inter-species 18S DNA differentiation and a very high taxonomic distance to the reference strain *D. tertiolecta*, respectively. In fact, for each strain, their closest relative has an 18S identity below 99.5% (Appendix A), which can be considered a new species. However, more taxonomic maker analysis has to be performed to adequately assign taxonomic relatedness, which was not the focus of this study.

Even though all isolates belong to the same genus, they also differ in their phenotypical appearance. While some species only show a size of 5 µm, others grow up to a diameter of 15 µm, such as *D. tertiolecta* (Figure 3).

Figure 4 demonstrates that after 14 days of cultivation, the OD_750_ of the different strains ranged from 1.6 to 3.8. In addition, some strains, such as *D. tertiolecta*, already enter the stationary phase after 8 days, while other strains, such as *Dunaliella* sp. 96 and *Dunaliella* sp. 37, continue growing linearly by day 14. 

The isolates not only varied in size and OD_750_ but also differed in their glycerol biosynthesis potential. The reference strain, *D. tertiolecta*, was used to investigate whether glycerol is produced intracellularly only or if the cells also secrete glycerol into the medium, as published by Chow et al. [9]. In the present study, glycerol was found only inside the cell, and nothing was present in the surrounding medium (Appendix A). These results are in alignment with Ben-Amotz et al., who described that the major function of glycerol is to maintain the osmotic balance [60]. As no glycerol was found in the medium, only the intracellular glycerol was analyzed in the subsequent experiments. 

In particular, *Dunaliella* sp. 96 was identified as a promising candidate for future investigations with a glycerol concentration of 0.36 mg mL^−1^, which is almost twice as much as *D. tertiolecta* (Figure 4b). Therefore, this isolate and the reference strain, *D. tertiolecta*, were chosen for the experimental optimization of the glycerol titer. The optimized conditions identified in this study for the generation of high glycerol titers were applied to the four most promising candidates: *D. tertiolecta*, *Dunaliella* sp. 27, *Dunaliella* sp. 83, and *Dunaliella* sp. 96.

### 3.1. Effect of Salinity

Ben-Amotz et al. found that the accumulated glycerol correlates with the extracellular salt concentration [8,10]. Accordingly, the algae were cultivated with different NaCl concentrations, and after two weeks, the NaCl concentration was increased either to 2 M, 3 M, or 4 M. 

The aim of this study was to increase the total glycerol concentration. It has already been reported that a too high increase in salt concentration leads to cell bursting [61,62], and the final salt concentration we aimed to reach was 2 M, 3 M, and 4 M. Therefore, it was decided that the best procedure to study the impact of salt was to cultivate the algae in medium containing 1 M, 1.5 M, and 2 M NaCl, which is in the range of optimal salt concentration for *Dunaliella* growth [6].

Both *Dunaliella* strains grew best in culture medium added with 1 M NaCl. After 14 days *D. tertiolecta* cells reached only an OD_750_ of 1.52 in 1.5 M NaCl and 1.39 in 2 M NaCl, compared to 1.66 in the control, cultivated in 1 M NaCl. *Dunaliella* sp. 96 reached an OD_750_ of 3.0 at 1 M NaCl, which is nearly two times the corresponding OD_750_ of the *D. tertiolecta* reference. Interestingly, *Dunaliella* sp. 96 appears to be quite sensitive to an increase in salt concentration, as the cells grown in 1.5 M and 2 M NaCl only reached ~80% and ~65% of the value for 1 M NaCl. This growth variation was manifested not only by a lower biomass formation (Figure 5), but also by a profound change in morphology, i.e., cellular aggregation and clumping as observed in Appendix A.

*D. tertiolecta*’s highest glycerol titer was achieved when cells were cultured in 1.5 M or 2 M NaCl, followed by an increase to 3 M NaCl, but also if cells were cultured in 2 M NaCl with a hyperosmotic change to 4 M NaCl (Figure 6). The resulting extracts all provided a glycerol concentration of 0.26 mg mL^−1^, increasing the glycerol biosynthesis by 14%. Cells grown in 1 M or 1.5 M did not survive the sudden NaCl increase to 4 M, resulting in lysed cells and almost no extracted glycerol. The cells grown in 1 M NaCl followed by an increase to 3 M NaCl have a high standard deviation, which indicates that some cells are already lysed while others could grow under hyperosmotic conditions (Figure 6). This high standard deviation is also visible for *Dunaliella* sp. 96. 

The highest glycerol concentration extracted from *Dunaliella* sp. 96 was reached when cells were grown at 1.5 M NaCl, followed by an increase to 3 M or 4 M NaCl. These conditions increased the glycerol yield by 19% to 0.51 mg mL^−1^. Interestingly, although the cells formed clumps when culturing at 1.5 M and 2 M NaCl and had a lower OD_750_ compared to the ones grown in 1 M NaCl, these cells still survived the hyperosmotic change and were able to adapt to the new NaCl concentration through glycerol production. The adapted cells are depicted in Figure 7. When cells were shocked with NaCl from 2 M to 4 M, the cells formed more compact clumps. By contrast, when cells were shocked with NaCl from 1.5 M to 4 M, clumps were less compact, but cells appeared to be less vital. However, even though the formed cell clumps differ in their phenotype after the hyperosmotic change to 4 M, the cells of both conditions survived the sudden salt increase.

In summary, the best condition for the NaCl shock appears to be the transition from 1.5 M to 3 M. Even though other conditions resulted in same increase in extracted glycerol, higher salt load, such as 4 M NaCl, should be avoided, as salt impedes production processes and causes corrosion of steal-containing photobioreactor parts at industrial scale [63,64]. 

### 3.2. Effects of Light Intensity and Wavelength

Since the extracted glycerol correlates with biomass formation, we not only investigated parameters that directly influence glycerol biosynthesis but also parameters improving the algae growth pattern. Consequently, the impact of light on algae growth was evaluated. Accordingly, the algae were cultivated with different light intensities (100, 500, 1000, 1500, and 2000 μmol m^−2^ s^−1^) and wavelengths, specifically white, red, and blue light (sunlight spectrum AM1.5 G for white light, 680 nm and 740 nm for red light, as well as 425 nm, 455 nm, and 470 nm for blue light). 

Interestingly, light intensity, but not color, had the highest impact on algae growth. With an illumination of white light at 500 μmol m^−2^ s^−1^, the growth of the *D. tertiolecta* increased by ~150%, while at 1000 μmol m^−2^ s^−1^ it increased by ~250% compared to the 100 μmol m^−2^ s^−1^ reference control with OD_750_ = 1.66 (Figure 8). Further, the increase to 1500 and 2000 μmol m^−2^ s^−1^ only resulted in slight growth improvements. 

Moreover, the growth of *Dunaliella* sp. 96 doubled to OD_750_ = 6.4 by an increase in white light intensity from 100 to 500 μmol m^−2^ s^−1^, and was even 2.5 times higher (OD_750_ = 7.8) when light intensity was further increased to 1000 μmol m^−2^ s^−1^. This cell growth could not be improved anymore by higher light intensities. Similar to cells of *Dunaliella* sp. 96, grown in 1.5 M and 2 M NaCl, the cells cultivated at light intensities higher than 500 μmol m^−2^ s^−1^ also formed clumps. Starting at the fifth day of cultivation at 2000 μmol m^−2^ s^−1^, the higher the light intensity, the earlier the clump formation started. The size of the clumps increased with increasing light intensity as well as with higher cell densities. This indicates that illumination with intensities higher than 500 μmol m^−2^ s^−1^ implies significant stress for the cells. 

The red and blue light impacted the growth rate of *Dunaliella* sp. 96 similar to *D. tertiolecta*, as the resulting OD_750_ was significantly lower compared to the white light at 1000 μmol m^−2^ s^−1^. Interestingly, for both strains at 100 μmol m^−2^ s^−1^, red light resulted in the highest cell growth. However, cells grew best with white light when intensities were raised to 1000 μmol m^−2^ s^−1^.

Figure 9 depicts that the cultivation at 1000 μmol m^−2^ s^−1^ white light causes higher glycerol titers for *Dunaliella* sp. 96 (0.53 mg mL^−1^), compared to red (0.23 mg mL^−1^) and blue light (0.45 mg mL^−1^), respectively. The glycerol extracted from *D. tertiolecta* does not show such a variation (white: 0.49 mg mL^−1^, red: 0.37 mg mL^−1^, and blue 0.40 mg mL^−1^).

At 100 μmol m^−2^ s^−1^, there is no significant difference between the light colors, with ~0.20 mg mL^−1^ for *D. tertiolecta* and 0.29 mg mL^−1^ for *Dunaliella* sp. 96. The glycerol biosynthesis attains saturation for light intensities higher than 500 μmol m^−2^ s^−1^, with ca. 0.45 mg mL^−1^ extracted from *D. tertiolecta* and 0.55 mg mL^−1^ from *Dunaliella* sp. 96. Thereby, both strains attained double the amount of extracted glycerol compared to the control. Even though higher biomass was achieved with light intensities above 1000 μmol m^−2^ s^−1^ compared to 500 μmol m^−2^ s^−1^, the higher biomass did not result in a higher glycerol titer. This again indicates that light intensities higher than 500 μmol m^−2^ s^−1^ imply a high level of stress for the cells. 

### 3.3. Combination of Identified Conditions for Increased Glycerol Yield

Light intensity was identified as having the highest impact on growth and glycerol biosynthesis. Subsequently, it was evaluated if the high light intensity alone leads to the highest glycerol concentration or if the combination with optimized salt conditions synergistically further increases glycerol yield. Optimal light conditions were identified as white light of 500 µmol m^−2^ s^−1^. This was combined with optimal salt concentration, i.e., a sudden increase in salt concentration from 1.5 M to 3 M after a growth phase of 14 days instead of a change from 1 M to 2 M. Even though the optimization was developed with *D. tertiolecta* and *Dunaliella* sp. 96, the four most promising *Dunaliella* strains according to Figure 4 (*D. tertiolecta*, *Dunaliella* sp. 96, *Dunaliella* sp. 27, and *Dunaliella* sp. 83) were also cultivated at these optimized conditions.

The algae growth under light-optimized conditions was increased by ~30% for *Dunaliella* sp. 27, ~75% for *Dunaliella* sp. 83, ~65% for *Dunaliella* sp. 96, and even ~215% for *D. tertiolecta* compared to the non-optimized conditions (Figure 10a). The highest OD_750_ of 5.04 was reached by *Dunaliella* sp. 96 when cultured at 500 µmol m^−2^ s^−1^ and 1 M NaCl, followed by *Dunaliella* sp. 27 at light-optimized conditions, as well as with light- and salt-optimized conditions. While for *Dunaliella* sp. 27 the growth was almost the same under light-optimized as well as light- and salt-optimized conditions, the growth of the other algae was decreased when cultured in 1.5 M NaCl compared to 1 M NaCl under optimized light conditions. The most significant reduction occurred with *Dunaliella* sp. 83 and 96, showing a decrease of approximately 18%, whereas the growth of *D. tertiolecta* decreased by ~9%. 

After glycerol extraction, the combined optimization of light and salt led to the highest glycerol concentration in all four tested *Dunaliella* strains. The most noticeable enhancement from solely optimizing light to the combined optimization of light and salt was observed with *Dunaliella* sp. 27 and *Dunaliella* sp. 96. Even though, under light- and salt-optimized conditions, *Dunaliella* sp. 27 reached the highest OD_750_, it did not lead to the highest extracted glycerol and resulted in 0.76 mg mL^−1^ only. This concentration is quite similar to the concentration of the extract from the reference strain *D. tertiolecta* (0.79 mg mL^−1^) and *Dunaliella* sp. 83 (0.74 mg mL^−1^). In contrast, the extracted glycerol of *Dunaliella* sp. 96 was ~20% higher in comparison to the other strains, resulting in a glycerol yield of 0.94 mg mL^−1^. 

Thus, with optimized conditions, only *Dunaliella* sp. 96 showed a significantly higher glycerol biosynthesis compared to the final value obtained with the reference strain *D. tertiolecta*. However, as the conditions were optimized for *Dunaliella* sp. 96 and *D. tertiolecta*, in future studies, the other two isolates, *Dunaliella* sp. 27 and *Dunaliella* sp. 83, should be optimized individually to increase their extracted glycerol titer potentially further. 

The NMR analysis shown in Figure 11 indicates that the extract isolated from algae consists of glycerol, when compared to the reference ^1^H-NMR spectra of glycerol [65]. Additionally, there are some other peaks visible, indicating impurities in the sample. As proteins are almost insoluble in ethanol [66], it is unlikely that proteins are present. The impurities might consist of cell components or carbohydrates, as these are slightly soluble in ethanol [67].

## 4. Discussion

The new isolates used in this study were identified as *Dunaliella*. However, it is challenging to identify the exact species using the 18S, ITS, and rubisco gene sequences, as various *Dunaliella* species were suggested with the same identity. This problematic identification is in line with the results from Highfield et al. and Assunção et al., who describe the diversity of the *Dunaliella* strains [1,2]. Using six different primer pairs, Highfield et al. classified the isolates in their study into one of four sub-clades, but they were not able to determine the exact species. In their opinion, the classification of *Dunaliella* has presented a challenge for many decades, and it is necessary that the scientific community tackle this issue to facilitate the identification of their working strains. The exact identity of a strain would enable targeted selection for specific applications [2].

As phylogenetic analysis of the 18S, ITS, and rubisco regions did not enable an exact identification of the *Dunaliella* strains, a taxonomic analysis was performed, and a phylogenetic tree was constructed to assess the relationships among the strains. The phylogenetic tree of the strains reveals (Figure 2) that all isolated *Dunaliella* strains are not only more distantly related to our reference strain, *D. tertiolecta*, than compared to each other, but also to any known *Dunaliella* strain, including the most characterized, *D. salina*. *D. tertiolecta* UVEX 999 was obtained in Oslofjord, Norway, while the isolates were collected in Australia, more than 14,000 km away from Norway. Consequently, the geographical distance from *D. tertiolecta* UVEX 999 is high, which could account for the increased phylogenetic distance. Interestingly, strains that were collected from the same lake, such as *Dunaliella* sp. 27 and *Dunaliella* sp. 37 are not immediately related. Comparing only the isolates, *Dunaliella* sp. 27 is the strain that shows the highest taxonomic difference among already known strains. 

The genetic differences to *D. tertiolecta* make it worth analyzing factors such as light and salt again, as the effects of these factors might differ from those tested with other *Dunaliella* strains. Even though the isolates all belong to the genus *Dunaliella*, their cell sizes range from 5 to 15 µm under the same cultivation conditions. Furthermore, the final OD_750_ of the isolates varied significantly between 1.6 (*D. tertiolecta*) and 3.8 (*Dunaliella* sp. 27), which under identical growth conditions indicates differential metabolic and photosynthetic rates. These will be assessed by a comprehensive system biology approach in further studies [68]. The difference in size and growth between *Dunaliella* strains has already been observed by Xu et. al. and Borowitzka et al. [5,52], who described a range of 7 µm to 12.5 µm and a range of 5 µm to 20 µm, respectively. Additionally, the specific growth rate in their study ranged from 0.1 to 0.7 per day at 200 μmol m^−2^ s^−1^ [52]. These results highlight the diversity within the *Dunaliella* clades and underscore the requirement for individual investigation of each isolate. 

At the measured time point, glycerol was only accumulated inside the cell and was not excreted into the medium, in contrast to the earlier findings of Chow et al. [9] and Tan et al. [69]. Due to the differences within the *Dunaliella* genus, it may be assumed that their strain secretes glycerol, while our strains only accumulate glycerol inside the cell. The results of Tan et al. have shown that extracellular glycerol occurred after 8 days and increased over time. Therefore, it might be possible that, in our study, no extracellular glycerol was observed in the short-term (within 24 h after salt stress) but will be present at later time points. If our strains also secrete glycerol during later stages of cultivation, should be analyzed in future studies. Glycerol secretion could facilitate the industrial-scale production of algae-based glycerol by enabling continuous cultivation [70]. 

### 4.1. Effect of Salt Concentration

The cultivation with different salt concentrations and hyperosmotic changes to various salt concentrations should increase accumulated glycerol. Our results show that the algae grew best in 1 M NaCl, with less growth in 1.5 M and even less in 2 M, as shown in Figure 5. The decreased growth with increasing salt concentration was more evident with *Dunaliella* sp. 96 than with the reference strain *D. tertiolecta*. While the growth of *D. tertiolecta* was decreased by 8% in 1.5 M NaCl and 17% in 2 M NaCl compared to 1 M NaCl, the growth of *Dunaliella* sp. 96 was lowered by ~35% when cultured in 2 M NaCl. Shariati et al. and Borowitzka et al. [8,59,71] also discovered varying growth of *Dunaliella* strains at different salinities. Additionally, Xu et al., e.g., describe 0.5 M as the optimal NaCl salinity for one strain, while other strains grow better at 1.5 M [52,72]. This again highlights the differences between the *Dunaliella* strains, which appear to be highly adapted to their environmental conditions, and the need to examine each strain individually. The cultivation of *Dunaliella* sp. 96 at a higher salt concentration forced the cells to accumulate and form clumps with each other. These clumps have already been characterized as a palmella stage of the cells [5,73]. These authors discovered that cells in the palmella stage lose their eyespot and flagella and become more circular. Additionally, they reported that the cells excrete a slime layer, which allows them to divide repeatedly and form aggregations of green cells [5]. Algae enter the palmella stage when they are exposed to extreme conditions, such as a decrease or increase in salinity or high light intensities [5]. In this study, the higher the salt concentration, the earlier the cells entered the palmella stage and the bigger the clumps (Appendix A). This confirms the findings from Montoya et al. [74], who observed increasing salt concentration as a catalyst for the palmella structure formation of *Dunaliella* cells. 

Furthermore, our data indicate that extreme NaCl increases, such as 1 M to 4 M NaCl, lead to cell lysis. This is in alignment with Avron and Ben-Amotz [61,62], who discovered that *Dunaliella* cells are able to physically withstand three- to four-fold increases in salt concentration, while further hyperosmotic stress leads to lysis of the cells. To avoid cell-bursting, a gradual increase in the NaCl concentration may help the cells better adapt and should be investigated in future experiments. 

The cells of *Dunaliella* sp. 96 entered the palmella stage when they were cultured at 1.5 M and 2 M NaCl. Even though the palmella stage is a stress signal, they were able to survive the hyperosmotic change to 4 M NaCl. It is conceivable that the palmella stage enabled survival because the cells on the outside of the clumps protected the cells on the inside.

The highest glycerol concentration for both strains was obtained when cells were cultured at 1.5 M NaCl, followed by a doubling of NaCl concentration. Extracts from *D. tertiolecta* also resulted in the same glycerol titer when cells were cultured in 2 M NaCl, followed by a hyperosmotic change to 3 M or 4 M NaCl. However, these two conditions did not lead to the highest extracted glycerol concentration from *Dunaliella* sp. 96 but resulted in a concentration in the range of the control. In contrast, *Dunaliella* sp. 96 cultivation in 1.5 M NaCl followed by an increase to 4 M NaCl resulted in a higher glycerol titer compared to the control, while the same condition led to bursting of *D. tertiolecta* cells. This again indicates that cells behave differently to hyperosmotic changes and highlights the necessity of investigating each strain separately. Since the accumulated glycerol for both strains was improved, when cells were cultured with 1.5 M NaCl followed by a hyperosmotic change to 3 M NaCl, this condition was chosen to be the best condition.

### 4.2. Effect of Light Color and Intensity

By investigating the light color and intensity, we aimed to improve the growth pattern of the algae. Our results show that the growth of the algae increased with increasing light intensity, highlighting that, compared to salt concentration and light color, light intensity has the most significant influence on both algae growth and glycerol biosynthesis. This phenomenon was detectable up to an illumination of 1000 μmol m^−2^ s^−1^. A further increase to 1500 or 2000 μmol m^−2^ s^−1^ did not immediately lead to increased growth. Illumination with an intensity higher than 1000 μmol m^−2^ s^−1^ may cause overexcitation of the photosystems, which leads to the formation of reactive oxygen species as a by-product. Reactive oxygen species cause irreversible photo-oxidative damage if not intercepted by the cell [75], resulting in stunted growth. Interestingly, the results of Sui et al. [76] show that the growth of their *D. salina* could only be increased up to an intensity of 200 μmol m^−2^ s^−1^, assuming that different *Dunaliella* strains adapt to different requirements of their environmental habitat. 

Algae have two photosystems (PS), which are responsible for the photosynthetic fixation of CO_2_ and are specialized for different sections of the light spectrum. The main function of PS I is to form NADPH, while PS II hydrolyzes water and synthesizes ATP. In contrast to PS II, which is primarily activated by blue light, PS I is more sensitive to (far-) red light [77]. In our experiments, the algae grew best with red light at low light intensity (100 μmol m^−2^ s^−1^), which suggests that PS I might play a bigger role in the photosynthetic activity at low light intensities. However, this requires a more detailed investigation. Our results also align with the work of Zhao et al. [78] who discovered that red LED produced the highest cell number of *Chlorella*. 

It is well known that blue and red light promotes optimal algae and plant cultivation because of the corresponding peaks in the absorption spectrum [79,80,81]. In our experiments, however, blue illumination resulted in the least biomass, independent of light intensity. Although some plants exhibit improved photosynthesis under blue than under red light [82], the study of Paper et al. [83] has demonstrated that blue light results in approximately half the maximal growth and biomass formation when compared to white and red illumination conditions. 

Compared to low light intensity, algae grown at higher light intensity (1000 μmol m^−2^ s^−1^) resulted in the highest biomass when illuminated with white light. This suggests that the composition of the light impacts the photosynthetic activity. Red or blue light on its own may cause an imbalance in the stimulation of the two photosystems and in the electron chain [84]. This imbalance may produce reactive oxygen species and photo oxidative damage, both of which are harmful to the organism [77,85]. 

The highest biomass production was achieved with white light illumination, indicating, that algae improved their photosystems to optimally function under white, sun-like light conditions [81]. As the white light spectrum is preferred, potential large-scale production may be realized in open ponds with sunlight exposure. Although algae’s glycerol production has not reached industrial scale, *D. salina* is already cultivated in an open pond system for β-carotene production [86]. In that respect, glycerol production can be realized through a similar process.

Xu et al. and Harvey et al. [52,86] reported that their *Dunaliella* strains turned red and produced β-carotene when cultured at high intensities of light or at red light. However, when cultured at light intensities higher than 500 μmol m^−2^ s^−1^, our strains remained green and did not turn red. Additionally, instead of β-carotene production, isolate *Dunaliella* sp. 96 entered the palmella stage. As already mentioned, algae enter the palmella stage when they are exposed to extreme conditions [5,73]. This indicates that the algae were stressed by light intensities higher than 500 μmol m^−2^ s^−1^ and consequently did not increase the overall glycerol titer, although they produced more biomass. Moreover, the cumulative data indicate that *Dunaliella* species either generate pigments or glycerol as a stress response.

Since illumination with 500, 1000, 1500, and 2000 μmol m^−2^ s^−1^ all resulted in approximately the same glycerol concentration, white light with an intensity of 500 μmol m^−2^ s^−1^ was identified as optimal illumination to avoid photo oxidative stress reactions in the cells. If algae-based glycerol production were to be operated on a large scale, the intensity of 500 μmol m^−2^ s^−1^ would reduce cooling needs because it consumes less energy and produces less heat. 

### 4.3. Optimized Light and Salt Conditions

Finally, it was evaluated if the combination of optimized salt concentration and light regime caused a synergistic beneficial effect on the glycerol yield. Light- and salt-optimized condition implies the cultivation at white light with an intensity of 500 μmol m^−2^ s^−1^ in 1.5 M NaCl, followed by an increase to 3 M NaCl after 14 days. While light-optimized cultures were cultivated at white light with an intensity of 500 μmol m^−2^ s^−1^ in 1 M NaCl and a sudden increase to 2 M after 14 days. These conditions were applied to the already investigated strains *D. tertiolecta* and *Dunaliella* sp. 96, as well as to the isolates *Dunaliella* sp. 27 and *Dunaliella* sp. 83, which were also promising candidates according to Figure 4. 

With the light- and salt-optimized condition, only *Dunaliella* sp. 96 showed a higher glycerol concentration (0.94 mg mL^−1^) compared to the reference strain *D. tertiolecta* (0.79 mg mL^−1^). As *Dunaliella* sp. 27 with the highest OD_750_ under light- and salt-optimized condition did not lead to the highest extracted glycerol, the growth pattern of the algae might be decoupled from the glycerol biosynthesis. To further analyze the glycerol accumulation potential of each strain individually, the glycerol concentration per cell was calculated for the non-optimized and the light- and salt-optimized condition (Figure 12). 

As for the hyperosmotic change, the NaCl concentration was doubled, it was excepted, that the intracellularly accumulated glycerol should double as well. However, the doubled amount of intracellularly accumulated glycerol was only seen for the non-optimized condition (~1.94-fold) (*Dunaliella* sp. 27: 0 h: 9.4 pg cell^−1^, 24 h: 21.4 pg cell^−1^. *Dunaliella* sp. 83: 0 h: 36.2 pg cell^−1^, 24 h: 51.2 pg cell^−1^. *Dunaliella* sp. 96: 0 h: 30.0 pg cell^−1^, 24 h: 64.8 pg cell^−1^. *D. tertiolecta*: 0 h: 46.9 pg cell^−1^, 24 h: 89.6 pg cell^−1^), while for the optimized condition, the intracellularly accumulated glycerol was increased on average by 1.65 only. This indicates a ceiling saturation in the cellular glycerol content. Furthermore, as the increase in light intensity from 500 to 1000 μmol m^−2^ s^−1^ led to higher growth but to the same amount of extracted glycerol, it again might be assumed that too high light intensities lead to photoinhibition of the cell and thereby to reduced glycerol accumulation.

Since *Dunaliella* sp. 27 is the smallest of the four examined *Dunaliella* strains in this study, expectantly the glycerol amount per cell is the lowest (max. 16 pg cell^−1^) and achieves approximately only one fourth of *D. tertiolecta*’s accumulated glycerol (max. 59 pg cell^−1^). The accumulated glycerol per cell of *Dunaliella* sp. 83 and *Dunaliella* sp. 96, the size of which is between *Dunaliella* sp. 27 and *D. tertiolecta*, reached 71% and 75% compared to *D. tertiolecta*, respectively. These results are in alignment with Xu et al.’s [52], who found that algae accumulated between 25 and 200 pg glycerol per cell when cultured at 1.5 M. However, the alga that accumulated up to 200 pg cell^−1^ showed the slowest growth rate in their study and thus does not provide a suitable alternative to our isolate. Similar outcomes are found in our study, where *D. tertiolecta*’s cells accumulate the highest levels of glycerol while displaying the lowest OD (Figure 11a). Interestingly, the cells of *Dunaliella* sp. 27 and *Dunaliella* sp. 83 accumulate an equal amount of glycerol at 0 h and optimized conditions (1.5 M), as well as after 24 h of non-optimized condition (2 M). In contrast, *D. tertiolecta* and the isolate *Dunaliella* sp. 96 accumulate higher amounts of glycerol at 2 M NaCl (24 h—non-optimized), as assumed. These differences indicate the need for system biological analysis of the algae to further understand the mechanism by which *Dunaliella* cells adapt to hyperosmotic changes at the cellular level [68].

The algae-based glycerol can find application in the manufacturing of high-value products, e.g., in the food, pharmaceutical, and cosmetic sectors [11,12,13,14]. The obtained glycerol can also be utilized in the chemical industry, contributing to the production of various substances, such as propane-1,3-diol [87], propane-1,2-diol [18], acrolein [88], and allyl alcohol [89]. The research of Melcher et al. [90] has already discussed the conversion of algae-based glycerol to allyl alcohol as a potential intermediate during the ammoxidation of propylene to acrylonitrile [91]. Consequently, allyl alcohol can serve as a substrate to produce bio-derived acrylonitrile [92] and/or acrylic acid [93]. 

## 5. Conclusions

Seven new algae strains from different environmental sites were genetically identified as *Dunaliella* strains. The phylogenetic tree of the strains reveals that all isolated *Dunaliella* strains are more distantly related to *D. tertiolecta* than compared to each other. The inconsistent physiological and phenotypical features of the *Dunaliella* isolates in this and other studies call for a consolidated effort by the scientific community to solve this issue. 

The isolates capacity to generate glycerol compared to the reference strain, *D. tertiolecta,* was examined in this study. Experimental variation of salt concentration and light intensities showed potential to improve overall glycerol titer by the reference *D. tertiolecta* and the proprietary isolate *Dunaliella* sp. 96, with light intensity having the highest impact on growth and glycerol accumulation. It has been demonstrated that optimized light intensity combined with optimized salt conditions further increases glycerol synthesis compared to optimized light conditions alone. Optimal light and salt conditions were identified with white light of 500 µmol m^−2^ s^−1^ and a sudden increase in salt concentration from 1.5 M to 3 M after a growth phase of 14 days. With these improved conditions, the glycerol concentration for *D. tertiolecta* could be doubled to 0.79 mg mL^−1^ in comparison to 100 μmol m^−2^ s^−1^ and an increase of NaCl concentration to 2 M NaCl. Glycerol titer in extracts from *Dunaliella* sp. 96 could be improved to even reach 0.94 mg mL^−1^. Consequently, the [58] overall glycerol titer could be increased 2.2-fold (*Dunaliella* sp. 96) and 2.6-fold (*D. tertiolecta*). The improved glycerol yield is beneficial for the potential industrial production of glycerol from *Dunaliella*, which allows for cultivation on wasteland without competition to food production. This study indicates that glycerol production for green chemicals has industrial potential if the correct cultivation conditions and strain selection are applied. Our current data set indicates that the new *Dunaliella* sp. 96 has the potential to be established as a new candidate for technical glycerol biosynthesis over the literature strain *D. tertiolecta*. The cumulative data generally call for more research into the biodiversity of algae strains. More specifically, there is a need for a detailed exploration of the phylogenetic relationships of the *Dunaliella* family using modern systems biology technologies and the technical exploitation of new algae isolates.

The biochemical bases and metabolic networks leading to glycerol formation under differently timed physiological cues should be examined using advanced systems biology tools [68]. This would allow the identification of metabolic bottlenecks in glycerol production, which could either be addressed by genetic engineering or process design interventions. Combining the synergistic power of genetics with process optimization, may lead to a techno-economically and ecologically viable glycerol production process. In this context, algae-based glycerol synthetized from atmospheric CO_2_ can be converted to ‘green chemicals’ to replace several fossil-based chemicals.

## Figures and Tables

**Figure 1 microorganisms-12-01318-f001:**
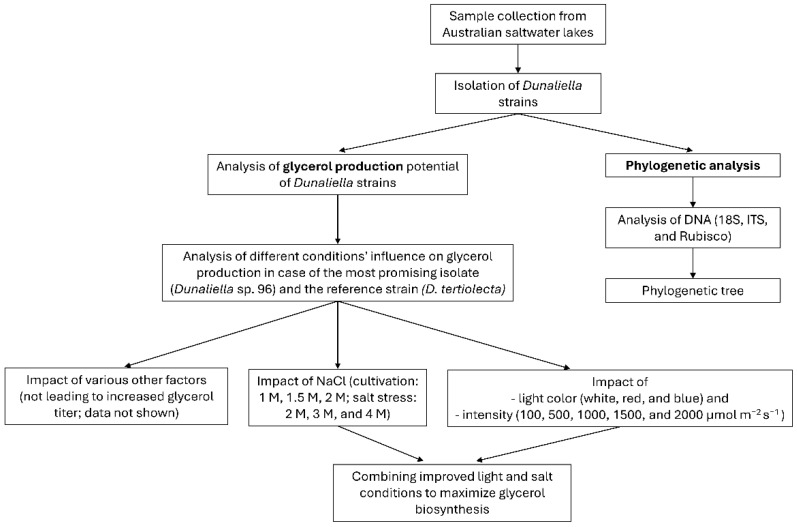
Flow chart of the conducted experiments during this study.

**Figure 2 microorganisms-12-01318-f002:**
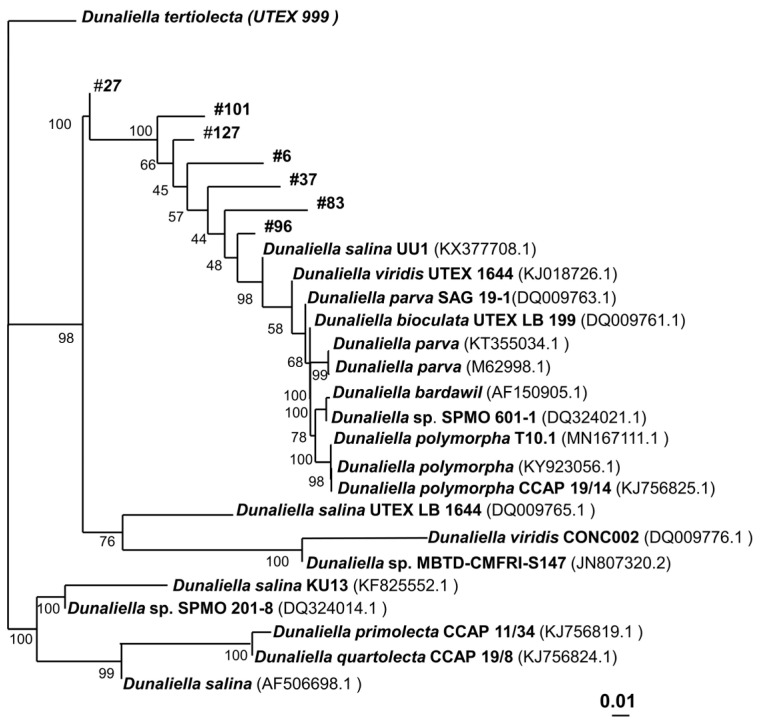
Phylogenetic dendrogram based on 18S rDNA gene sequence comparison indicating the position of *D. tertiolecta* and the new strains *Dunaliella* sp. 6–127 within the *Dunaliella* family, performed by the neighbor-joining method using software from PHYLIP, version 3.57c; the DNADIST program with Kimura-2 factor was used to compute the pairwise evolutionary distances for the above aligned sequences, the topology of the phylogenetic tree was evaluated by performing a bootstrap (algorithm version 3.6b) with 1000 bootstrapped trials. The tree was drawn using Tree View 32 software. Bar corresponds to 1 nucleotide substitution per 100 nucleotides.

**Figure 3 microorganisms-12-01318-f003:**
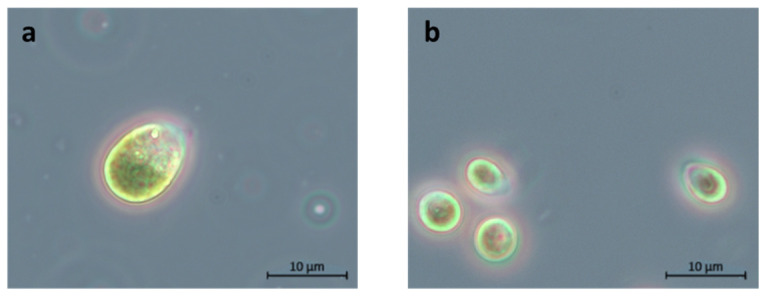
Bright field images (100×) of the morphology of the biggest and the smallest strain of *Dunaliella* analyzed in this study, cultivated at 1 M NaCl. (**a**) *Dunaliella tertiolecta* (UTEX 999). (**b**) *Dunaliella* sp. 127, an environmental isolate.

**Figure 4 microorganisms-12-01318-f004:**
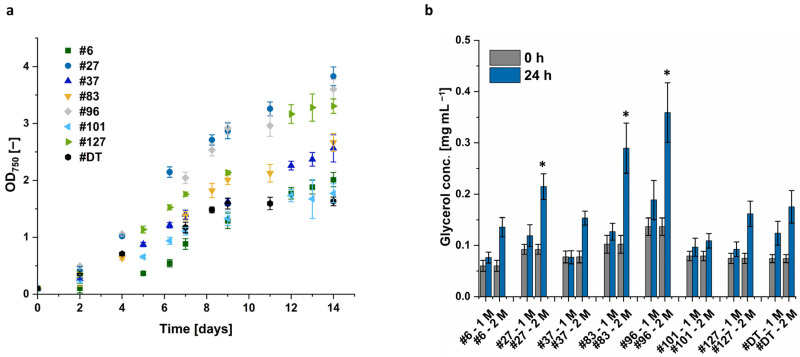
(**a**) Growth curve of eight different *Dunaliella* strains. #DT refers to *D. tertiolecta*. #6–#127 refers to the different *Dunaliella* isolates *Dunaliella* sp. 6–127. Algae were cultivated in 1 M NaCl containing modified Johnson medium with 100 μmol m^−2^ s^−1^ illumination for 14 days. (**b**) Glycerol titer of these eight *Dunaliella* strains after 14 days of cultivation (0 h) and 24 h after NaCl concentration was increased to 2 M NaCl or remained constant at 1 M NaCl. After collecting the biomass, glycerol was extracted. A statistical analysis of glycerol extracts of the isolates at 24 h compared to the corresponding control of *D. tertiolecta* was performed via *t*-test and the level of statistical significance (*) was set to *p* < 0.05. The error bars represent the standard deviation of at least triplicates.

**Figure 5 microorganisms-12-01318-f005:**
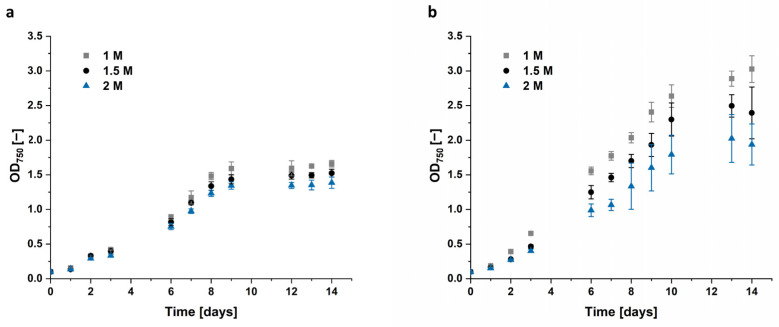
Growth curve of (**a**) *D. tertiolecta* and (**b**) *Dunaliella* sp. 96 cultivated in modified Johnson medium (pH = 7.5) at 28 °C with 100 μmol m^−2^ s^−1^ illumination for 14 days. NaCl concentration of the medium varied between 1 M, 1.5 M, and 2 M. The error bars represent the standard deviation of at least triplicates.

**Figure 6 microorganisms-12-01318-f006:**
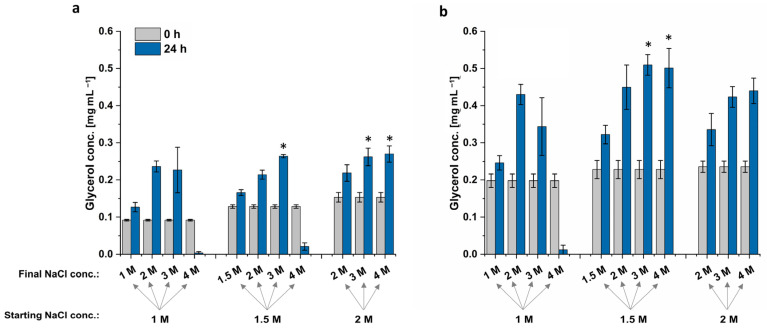
Glycerol concentration of different *Dunaliella* strains (**a**) *D. tertiolecta* and (**b**) *Dunaliella* sp. 96 cultivated 14 days in modified Johnson medium (pH = 7.5) at 28 °C with 1 M, 1.5 M or 2 M NaCl. After 14 days, NaCl concentration was increased to either 2 M, 3 M or 4 M NaCl or kept constant. Glycerol concentration was measured at time point 0 h and 24 h after hyperosmotic changes. A statistical analysis of differences between tested conditions at 24 h compared to the control (cultivated at 1 M followed by an increase to 2 M NaCl) was performed via *t*-test and the level of statistical significance (*) was set to *p* < 0.05. The error bars represent the standard deviation of at least triplicates.

**Figure 7 microorganisms-12-01318-f007:**
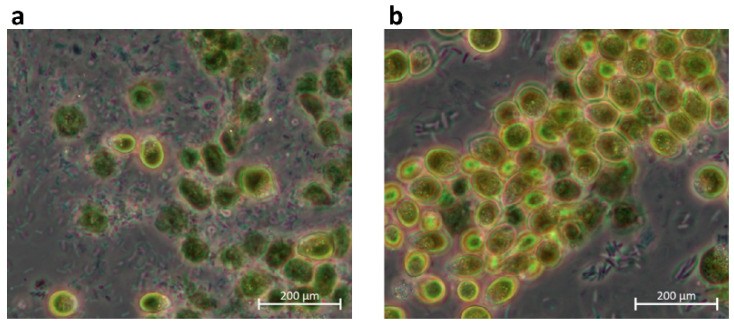
Bright field images (20×) of *Dunaliella* sp. 96 after hyperosmotic changes. Cells were cultivated with (**a**) 1.5 M or (**b**) 2 M NaCl for 14 days followed by a hyperosmotic change to 4 M NaCl.

**Figure 8 microorganisms-12-01318-f008:**
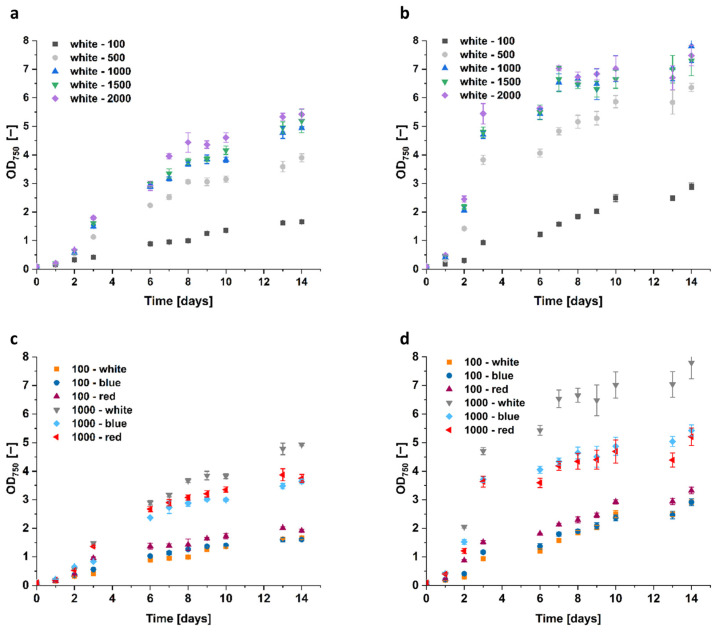
Growth curve of *Dunaliella* grown with different light intensities and colors. (**a**) *D. tertiolecta* and (**b**) *Dunaliella* sp. 96 were cultivated in modified Johnson medium (pH = 7.5) at 28 °C over 14 days with the illumination of white light at varying intensities (100–2000 μmol m^−2^ s^−1^). The impact of the illumination of white, red, and blue light with the intensity of 100 and 1000 μmol m^−2^ s^−1^, respectively, for (**c**) *D. tertiolecta* and (**d**) *Dunaliella* sp. 96 when cultured in modified Johnson medium (pH = 7.5) at 28 °C over 14 days are shown. The error bars represent the standard deviation of at least triplicates.

**Figure 9 microorganisms-12-01318-f009:**
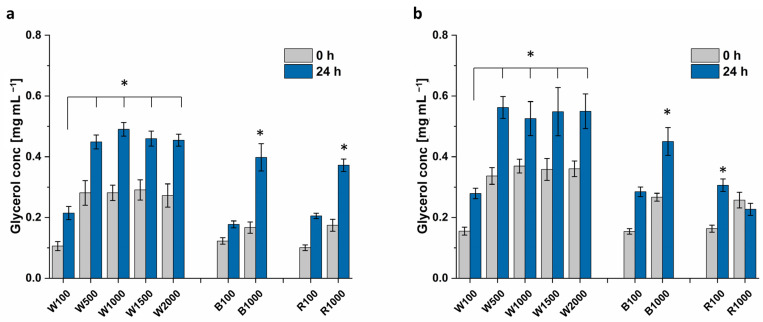
Glycerol concentration of (**a**) *D. tertiolecta* and (**b**) *Dunaliella* sp. 96. Algae were cultivated in 1 M NaCl containing modified Johnson medium (pH = 7.5) at 28 °C with different light intensities and colors. (W = white, B = blue, R = red. The numbers refer to the light intensity in μmol m^−2^ s^−1^). After 14 days of cultivation the NaCl concentration was increased to 2 M. Glycerol concentration was measured at time point 0 h and 24 h after variation of salt concentration. A statistical analysis of differences between tested conditions compared to the control (100 μmol m^−2^ s^−1^, white light, 24 h) was performed via *t*-test and the level of statistical significance (*) was set to *p* < 0.05. The error bars represent the standard deviation of at least triplicates.

**Figure 10 microorganisms-12-01318-f010:**
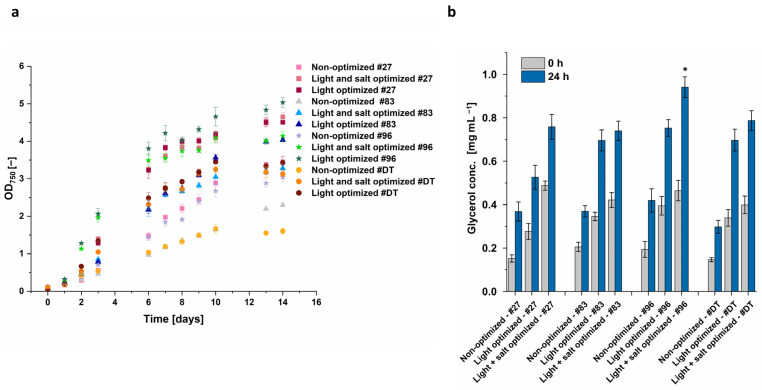
(**a**) Growth curve of *Dunaliella* sp. 27, *Dunaliella* sp. 83, and *Dunaliella* sp. 96 (#27, #83, and #96) and *D. tertiolecta* (#DT) over 14 days when cultivated under light-optimized conditions (500 μmol m^−2^ s^−1^ white light and 1.0 M NaCl modified Johnson medium, pH = 7.5), optimized conditions (500 μmol m^−2^ s^−1^ white light and 1.5 M NaCl modified Johnson medium, pH = 7.5) or under non-optimized conditions (100 μmol m^−2^ s^−1^ white light and 1 M NaCl modified Johnson medium, pH = 7.5) at 28 °C. (**b**) Glycerol concentration 0 h and 24 h after NaCl concentration was increased to 2 M (non-optimized and light-optimized) or 3 M (light and salt optimized). A statistical analysis of differences between the conditions compared to optimized conditions of *D. tertiolecta* at 24 h was performed via *t*-test and the level of statistical significance (*) was set to *p* < 0.05. The error bars represent the standard deviation of at least triplicates.

**Figure 11 microorganisms-12-01318-f011:**
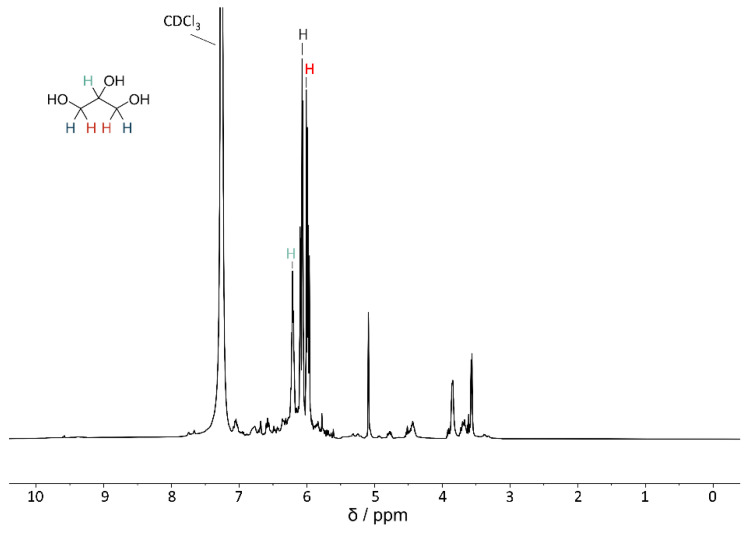
^1^H-NMR spectrum of the ethanol extracted glycerol from *D. tertiolecta* recorded on a Bruker Ascend 400 MHz spectrometer. All spectra are referenced on the proton signal of CDCl_3_ (7.26 ppm). The color-coded hydrogens in the glycerol structure correlate with the respective signals in the spectrum marked by the colored hydrogens.

**Figure 12 microorganisms-12-01318-f012:**
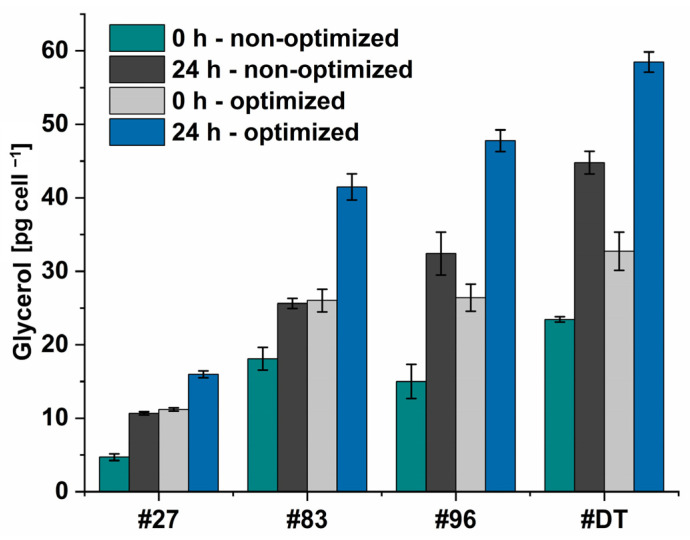
Glycerol concentration per cell for four different *Dunaliella* strains. #DT refers to *D. tertiolecta*, #27–#96 refers to *Dunaliella* sp. 27–96. Algae were cultivated under optimized condition (500 μmol m^−2^ s^−1^ white light and 1.5 M NaCl modified Johnson medium, pH = 7.5) or non-optimized condition (100 μmol m^−2^ s^−1^ white light and 1 M NaCl modified Johnson medium, pH = 7.5) at 28 °C. After 14 days of cultivation NaCl concentration was increased to 2 M (non-optimized) or 3 M (optimized) and glycerol was extracted at time points 0 h and 24 h. The corresponding amount of glycerol per cell was calculated for both time points.

## Data Availability

All data generated or analyzed during this study are included in this published article. However, if detailed values, are desired these data are available from the corresponding author on reasonable request. Additional information used to support the findings of this study are included within the Appendix A.

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
