# Peer review of "Evaluation of Chemical and Physical Triggers for Enhanced Photosynthetic Glycerol Production in Different Dunaliella Isolates"

_microorganisms, 2024, doi:10.3390/microorganisms12071318_

Round 1

Reviewer 1 Report

Comments and Suggestions for Authors

Reviewers comments on the article

Phylogenetic distribution and comparative glycerol production in various new Dunaliella isolates

General comments:

Line 14: “new to science” sounds to dramatic, please replace with newly discovered, or similar

Lines 18-19: “cultivation with white light of an intensity between 500 and 2000 μmol m−2 s−1 as opposed to 100 μmol m−2 s−1 achieves higher biomass...” this claim is true for the most green microalgal species and is not related to glycerol nor Dunaliella specifically. Please reformulate.

Lines 42-43: “a 2006 study discussed, that the taxonomy of the green algae genus Dunaliella is often perceived as confusing.” At first reading it seems that you found 2006 papers on the topic. On the second reading one realizes you refer to certain study from the year 2006. What is the study? Please reformulate this sentence.

Line 47: isolates' change to isolate's

Line 50: “grow with salt concentrations up to saturation” which is? Please state the numerical value

Line 89: Comment to the authors: If you grow your strains in urban wastewater medium for production of highly pure pharmaceutical grade glycerol, would it still be acceptable? If not please reformulate. Also the paragraph underestimates the problem of the abundant waste glycerol from biofuel production using edible oils.

Line 116: Is there any particulate reason you used D. tertiolecta UTEX 999 as the reference strain?

You claimed later that “D. tertiolecta was highly taxonomically different from the isolates, the phylogenetic tree only involves the seven isolates.”

Could let’s say D. salina be used, as a very well-known strain isolated from high salinity and saline waters and much closer phylogenetically to your isolates… Please explain in the text.

Line 132: what 16 sL h-1 means in term of gas flow? How many liters per liter of culture per min it is?

MgCl2•6H2O please use dot that is meant for this purpose like MgCl2·6H2O. The symbol you now use is meant for indexing

Also why did you use Johnson medium and not for example Guillard′s (F/2) marine medium? It is not clear what is the salinity of your medium. Is it prepared with demi water or artificial seawater? If you used demi water, then why do you cultivate marine halotolerant Dunaliella species in freshwater medium? Please explain and reformulate.

Line 188: “As the growth of algae is highly related to photosynthesis” this sentence is true for all autotrophic cultures in the world! Please reformulate

Line 216-220: Apart from triplicate measurements, how many biological replicates you had or how many flasks did you have for each condition. Please state clearly in the text.

Line 263: “with a glycerol concentration of 0.36 mg mL-1” If you only found glycerol inside the cell what this concentration means? Please explain clearly where did you measured this conc.

Line 286: data not shown - please avoid such statements and add these data to supplementary material

Figure 5a, the upper half of the figure is empty. Please change Y scale to OD750nm = 0.0 – 2.0

Line 293: “Both Dunaliella strains grew best in 1 M NaCl” you meant “Both Dunaliella strains grew best in culture medium added with 1 M NaCl”? Please reformulate.

Lines 296-298 “Dunaliella sp. 96 appears to be quite sensitive to an increase in salt concentration, as the cells grown in 1.5 M, and 2 M NaCl only reached ~80% and ~65% of the value for 1 M NaCl.”

Does this means Dunaliella sp. 96 is not halotolerant?

Please move Figure 6 to Appendix.

Figure 7a, the upper half of the figure is empty. Please rescale y axis.

Lines 353 and 356: color is technically and actually wavelength, I suggest to use these terms

Figure 9a and 9c, the upper half of the figure is empty. Please rescale y axis.

Lines 395-397: “Figure 10 depicts, that the cultivation at 1000 μmol m−2 s−1 white light causes higher glycerol titers (D. tertiolecta: 0.49 mg mL-1, Dunaliella sp. 96: 0.53 mg mL-1), compared to red (D. tertiolecta: 0.37 mg mL-1, Dunaliella sp. 96: 0.23 mg mL-1) and blue light (D. tertiolecta: 0.40 mg mL-1, Dunaliella sp. 96: 0.45 mg mL-1), respectively.”

From the Figure 10a I couldn’t conclude that glycerol was produced more in white light (500 – 2000) than in blue at 1000. Please reconsider your conclusions.

Although I understand the experimental strategy for the use of white, blue and red light, if You concluded that the optimal conditions are only achieved with white light - is there any real benefit to the readers to report all the data of the red and blue light experiments?

Line 495: Dunaliella sp. please use italic for Dunaliella

Line 508: “Tan et al. have shown that extracellular glycerol occurred after 8 days and increased over time” could this be because of the increasing cell rupture of the mature culture that releases intracellular glycerol into the culture medium? Please elaborate.

Line 529: Remove “with each other”. Also palmella stage is usually accompanied with change in color due to beta carotene accumulation? Have you noticed such biomass color change?

Line 590: Cyanobacteria has different photosynthetic apparatus than green microalgae and cannot be directly compared like that.

Lines 597-600: any PAR light above 170 kJ/mol of photons of energy and above light intensity that corresponds to compensation point of certain algae (approx. 30 µmol/m2/s) is able to maintain photosynthesis. Please remove this confusing section.

Lines 603-605: “As the white light spectrum is preferred, potential large-scale production may be realized in open ponds with sunlight exposure”

The summertime peak intensity of the sunlight is 1800 – 2000 µmol/m2/s between 10 am and 17 pm in many warm regions of the world. Your algae perform well only up to 500 µmol/m2/s. How do you plan to solve this issue?

Figure 13 shows that culture collection strain performs way better than any of the isolates. Does this recommend it to be used for glycerol production, instead?

Line 667: Please state the name of the strain you mention as the reference.

References: 23,27,33, 93 are incomplete.

Reference 50 is the PhD thesis, where it can be accessed? Please provide the link. If not available to the public, please remove.

References 54,55,59,60 are very old and should be replaced with more recent ones.

Please check your references and correct them when necessary.

Reviewer 2 Report

Comments and Suggestions for Authors

The submitted Manuscript provides interesting and novel data about glycerol accumulation by strains of Dunaliella. The Reviewer suggests to accept the Manuscript after revisions. The present Reviewer suggests major revisions to provide more time for Authors for corrections.

1) Figure S6, figure legend. The higher the salt concentration, the earlier the cells start to accumulate and the larger the formed palmella structures.

Not sure that the word “accumulate” fits here, probably “aggregate” or something better.

2) Abstract. Language. Word “report” is used too often. Pls, substitute by synonyms.

3) Abstract.

 Under 21 these optimal light intensity and salt conditions, the glycerol titer of D. tertiolecta could be doubled 22 to 0.79 mg mL-1 and was higher compared to singularly optimized conditions

Doubled compared to which conditions?

4) Figure 2. The Reviewer is puzzled by the fact that the tree is lacking Dunaliella tetriolecta which is widely described and studied by the Authors. A brief search leads to e.g. the paper https://thescipub.com/pdf/ajessp.2013.317.321.pdf

American Journal of Environmental Science 9 (4): 317-321, 2013 ISSN: 1553-345X ©2013 Science Publication doi:10.3844/ajessp.2013.317.321 Published Online 9 (4) 2013 (http://www.thescipub.com/ajes.toc) Corresponding Author: Duc Tran, School of Biotechnology, International University, Thu Duc Dist., VNU, Vietnam Science Publications 317 AJES Phylogenetic Study of Some Strains of Dunaliella

with Dunaliella tetriolecta close to D. parva based on ITS sequences.

Pls, indicate the location of D. tetriolecta at your figure 2 then or discuss the point more.

5) The Reviewer likes figure 13 very much. Could the Authors, please, add the concentrations/estimates of intracellular glycerol concentrations calculated per a cell and compare them with NaCl concentrations applied.

6) Methods.

After 14 days of cultivation the NaCl concentration was abruptly increased, if not 167 stated otherwise, to 2 M by adding the same volume of 3 M NaCl solution. As a control, 168 instead of 3 M the same volume of a 1 M NaCl solution was added. To measure the accu-169 mulated glycerol, the biomass was harvested by centrifugation of 1 mL culture

How long did it take between the NaCl increase and  the harvesting? It’s one of the main point to describe in methods.

7) PHYLIP, version 3.57c. The DNADIST program with Kimura-2 factor

Methods. Please, indicate the www sources for the software where appropriate with the dates accessed.

8) at different salt concentrations (1 M, 1.5 M, and 2 M) and after 14 182 days the NaCl concentration was raised to 2 M, 3 M, and 4 M by adding NaCl solutions 183 in the required molarity. As a control the cultures were diluted with the same volume, but 184 the salt concentration was retained constant.

Same as 6). What was the duration of high NaCl treatment?

9) Thus, 100 and 1000 μmol m−2 s−1 were adjusted 194 by using only light of the wavelengths 680 nm and 740 nm for red light conditions and 195 425 nm, 455 nm, and 470 nm for blue light.

What were the sources of the illumination under the conditions of red and blue light? The light energy was evidently about 1.5-2 times different for the conditions of red and blue light based on wavelength. How was the light intensity measured?

10) After 14 202 days of cultivation, salt was increased to 3 M NaCl. The non-optimized condition was 203 cultivated as mentioned before. 204

Same as 6 and 8. Duration of high NaCl treatment?

11) Figure 6, figure legend. the forming of accumulated cells.

Aggregated cells? Accumulated is not a good word here.

12) Figure 8. The quality of the figure is not extremely impressive. The Reviewer knows how difficult it is to get pictures of Dunaliella sp. at all but suggests to have a look if the better pictures could be found if available.

13) With these improved conditions, the glycerol 700 concentration for D. tertiolecta could be doubled to 0.79 mg mL-1 compared to former con-701 dition.

What are the former conditions? Pls, check the text for the issue.

14) Please, consider if reading/citing could be useful

Massjuk, N.P., 1973a. New taxons from the genus

Dunaliella Teod., I. Ukr. Bot. Zh., 30: 175-183.

Massjuk, N.P., 1973b. New taxons from the genus

Dunaliella Teod., II. Ukr. Bot. Zh., 30: 345-354.

15) Figure 8, looks like part b represents cells with more yellow tones probably due to higher carotene. Pls, consider discussing the point if it is so.

16) Decent and interesting Manuscript describing novel, scientifically and practically important results.

Comments on the Quality of English Language

Reasonable.

Reviewer 3 Report

Comments and Suggestions for Authors

In this study, the authors investigated the influence of increased salinity and the type and intensity of light on glycerol production in seven new isolates of Dunaliella and the reference strain D. tertiolecta. The results presented in this paper are interesting and may form the basis for further research in this area. However, there are some points that the authors should address.

The title of the paper is inappropriate considering that the paper aims to investigate the influence of NaCl and the intensity and type of light on the production of glycerol. In the paper, only the identification of isolates to the genus level was performed and a phylogenetic tree was presented. However, this was not the aim of the work, but only the accompanying information. Change the title according to the aim of the paper and the results presented.

Abstract

Lines 13-14 Delete are new to science and rephrase the sentence.

Introduction

Lines 103-106 In synergy to the obtained biochemical data, the new Dunaliella isolates were subjected to a rigorous phylogenetic analysis, which indicated that all strains were new to science and not related to the D. tertiolecta reference despite their capacity to biosynthesize glycerol in the absence of ß-carotene formation - Delete rigorous and new to science and rephrase the sentence.

Material and methods

Microalgae cultivation

Add how many days the cultivation lasted. Growth curves are shown in the results, but this part of the experiment is not mentioned anywhere in the methods.

Glycerol extraction and measurement

Lines 167-168 After 14 days of cultivation the NaCl concentration was abruptly increased, if not stated otherwise, to 2 M by adding the same volume of 3 M NaCl solution. - This sentence is completely unclear, rewrite it. Were the algae cultured in a modified Johnson medium with 1M NaCl and then the NaCl concentration was increased to 2M? How long were the algae grown after the NaCl concentration was increased, and were they grown under the same conditions as above? Was this done for all seven isolates?

Analyzing the impact of light on glycerol biosynthesis

Which strains were used in this part of the experiment? What medium was used here, with 1M NaCl?

Combining improved light and salt conditions to increase glycerol production titer

How long did the cultivation of the algae last after increasing the NaCl concentration and under what conditions was it done?

Results

Line 230-231 The analysis highlights that all seven isolates are new species, as their similarity to the closest relative falls below 99.5% (Table SI 4)- Delete this sentence because I am not sure if these isolates can be considered new species. Molecular tests to verify new species are more complex than the identification performed in this study.

Lines 273-275 Glycerol titer of these eight Dunaliella strains after 14 days of cultivation (0 h) and 24 h after NaCl concentration 274 was increased to 2 M NaCl or remained constant at 1 M NaCl.- The part of the experiment where the concentration of NaCl was increased to 2M, and in the second part, it remained at 1M, was not described in the materials and methods.

Figures 4. 10, 11-Shorten the text below the figures. There is too much information that has already been given in MM.

Lines 265-268  The optimized conditions dentified in this study for generation of high glycerol titers were applied to the four most promising candidates: D. tertiolecta, Dunaliella sp. 27, Dunaliella sp. 83, and Dunaliella sp. 96.- Optimization of glycerol production was conducted for all mentioned strains only in the final experiment, whereas isolate 96 and the reference isolate D. tertiolecta were used in all subsequent experiments. Explain this in the text.

Discussion

Figure 13 and all results related to Figure 13 should be transferred to the Results section.

Round 2

Reviewer 1 Report

Comments and Suggestions for Authors

I have no further comments.

Author Response

Thank you

Reviewer 2 Report

Comments and Suggestions for Authors

The Authors reasonably responded to the posed questions and improved the readability of the Manuscript. Some mostly minor points left are listed below.

1) Supplementary materials. It is better to number tables and figures separately: Table S1 etc. and then Figure S1 etc.

2) Figure 3 has the scale of 10 micrometers, figure 7 shows the scale of 200 micrometers for the aggregated cells, the same 200 micrometers are given at the present figure S7 (future figure S3?). Please, check if the scale of 200 micrometers is correct and add a few explanatory sentences then.

3) Figure S7, figure legend. Better “aggregate” or a better word, not “accumulate”.

4) Present figure 12. The rough estimate for Dunaliella tetriolecta gives intracellular glycerol concentrations about 1.3 M assuming cells of 15*10*10 micrometers (about 750 fL) and about 90 pg/cell. It fits the earlier publications (e.g. en-Amotz A, Avron M. The Role of Glycerol in the Osmotic Regulation of the Halophilic Alga Dunaliella parva. Plant Physiol. 1973 May;51(5):875-8).

However, figures 6 and present S5 indicate concentrations of 0.2 mg/ml or about 2 mM. Please, check if it is correct and also correct in the text.

Comments on the Quality of English Language

Minor changes are required to check  the correct numbers or misprints for sizes of cells and concentrations. 

Reviewer 3 Report

Comments and Suggestions for Authors

No comments

Author Response

Thank you